# Hepatotoxic Evaluation of *N*-(2-Hydroxyphenyl)-2-Propylpentanamide: A Novel Derivative of Valproic Acid for the Treatment of Cancer

**DOI:** 10.3390/molecules28176282

**Published:** 2023-08-28

**Authors:** Ana María Correa Basurto, Feliciano Tamay Cach, Rosa Adriana Jarillo Luna, Laura Cristina Cabrera Pérez, José Correa Basurto, Fernando García Dolores, Jessica Elena Mendieta Wejebe

**Affiliations:** 1Laboratorio de Biofísica y Biocatálisis, Sección de Estudios de Posgrado e Investigación, Escuela Superior de Medicina, Instituto Politécnico Nacional, Plan de San Luis y Salvador Díaz Mirón s/n, Casco de Santo Tomas, Ciudad de México 11340, Mexico; anmaricb235@gmail.com (A.M.C.B.); uva6210@yahoo.com.mx (L.C.C.P.); corrjose@gmail.com (J.C.B.); 2Laboratorio de Investigación de Bioquímica Aplicada, Sección de Estudios de Posgrado e Investigación, Escuela Superior de Medicina, Instituto Politécnico Nacional, Plan de San Luis y Salvador Díaz Mirón s/n, Casco de Santo Tomas, Ciudad de México 11340, Mexico; ftamay@ipn.mx; 3Laboratorio de Morfología, Sección de Estudios de Posgrado e Investigación, Escuela Superior de Medicina, Instituto Politécnico Nacional, Plan de San Luis y Salvador Díaz Mirón s/n, Casco de Santo Tomas, Ciudad de México 11340, Mexico; rajarillo@ipn.mx; 4Laboratorio de Farmacología, Departamento de Bioprocesos, Unidad Profesional Interdisciplinaria de Biotecnología, Instituto Politécnico Nacional, Avenida Acueducto s/n, La Laguna Ticoman, Ciudad de México 07340, Mexico; 5Laboratorio de Diseño y Desarrollo de Nuevos Fármacos e Innovación Biotecnológica, Sección de Estudios de Posgrado e Investigación, Escuela Superior de Medicina, Instituto Politécnico Nacional, Plan de San Luis y Salvador Díaz Mirón s/n, Casco de Santo Tomas, Ciudad de México 11340, Mexico; 6Laboratorio de Patología, Instituto de Ciencias Forenses de la Ciudad de México, Av. Niños Héroes 130. Col. Doctores, Delegación Cuauhtémoc, Ciudad de México 06720, Mexico; fernando.garcia@tsjcdmx.gob.mx

**Keywords:** acute oral toxicity, HO-AAVPA, liver injury

## Abstract

Valproic acid (VPA) is a drug that has various therapeutic applications; however, it has been associated with liver damage. Furthermore, it is interesting to propose new compounds derived from VPA as *N*-(2-hydroxyphenyl)-2-propylpentanamide (HO-AAVPA). The HO-AAVPA has better antiproliferative activity than the VPA in different cancer cell lines. The purpose of this study was to evaluate the liver injury of HO-AAVPA by acute treatment (once administration) and repeated doses for 7 days under intraperitoneal administration. The median lethal dose value (LD_50_) was determined in rats and mice (females and males) using OECD Guideline 425. In the study, male rats were randomly divided into 4 groups (n = 7), G1: control (without treatment), G2: vehicle, G3: VPA (500 mg/kg), and G4: HO-AAVPA (708 mg/kg, in equimolar ratio to VPA). Some biomarkers related to hepatotoxicity were evaluated. In addition, macroscopic and histological studies were performed. The LD_50_ value of HO-AAVPA was greater than 2000 mg/kg. Regarding macroscopy and biochemistry, the HO-AAVPA does not induce liver injury according to the measures of alanine aminotransferase, aspartate aminotransferase, alkaline phosphatase, glutathione peroxidase, glutathione reductase, and catalase activities. Comparing the treatment with HO-AAVPA and VPA did not show a significant difference with the control group, while malondialdehyde and glutathione-reduced levels in the group treated with HO-AAVPA were close to those of the control (*p* ≤ 0.05). The histological study shows that liver lesions caused by HO-AAVPA were less severe compared with VPA. Therefore, it is suggested that HO-AAVPA does not induce hepatotoxicity at therapeutic doses, considering that in the future it could be proposed as an antineoplastic drug.

## 1. Introduction

*N*-(2-hydroxyphenyl)-2-propylpentanamide (HO-AAVPA) is a recently designed valproic acid (VPA) derivative synthesized and evaluated in vitro as an antiproliferative compound in cancer cells [1]. The compound HO-AAVPA exhibits antiproliferative effects against various cancer cell lines, including cells of the cervical-uterine lineage (HeLa), rhabdomyosarcoma (A204), breast cancer (MCF-7, MDA-MB-231, and SKBr3), glioblastoma (U87-MG: human glioblastoma), and osteosarcoma (U-2 OS: human osteosarcoma), as well as antiangiogenic properties [2,3]. It has been suggested that HO-AAVPA induces antiproliferative effects through the inhibition of histone deacetylases (HDACs), which could be related to the translocation of the high mobility group box 1 protein (HMGB1) and the increase of the levels of reactive oxygen species (ROS), which could trigger cell apoptosis [3]. On the other hand, it has been shown that HO-AAVPA induces anticonvulsant effects comparable to those of VPA, coupled with the fact that HO-AAVPA presents lesser toxicity and damage to embryonic and fetal development than VPA [4]. Another study reports that fourth-generation poly(amidoamine) dendrimer (PAMAM-G4) was used as a drug delivery vehicle for HO-AAVPA in order to protect it from stress force degradation and maintain its antiproliferative effect in a breast cancer cell line (MCF-7). Also, PAMAM-G4 favored the water solubility of HO-AAVPA under basic and neutral pH conditions [5].

On the other hand, despite the fact that numerous studies have demonstrated the multiple therapeutic applications of VPA, including its evaluation as an antineoplastic in phase I and phase II trials [6,7,8], it is well known that VPA exerts different effects on liver function that modify various biochemical parameters, such as the increased ammonium (NH_4_^+^) and thiobarbituric acid reactive species (TBARS) levels and the decreased reduced glutathione (GSH) levels [9,10,11,12]. In addition, VPA causes microvesicular steatosis that can lead to fulminant hepatitis, idiopathic hepatitis, and hemorrhagic pancreatitis, and prenatal exposure induces autism spectrum disorder-like phenotypes in both humans and rodents, among other effects that can lead to death [12,13,14,15]. Part of these effects is attributed to the metabolism of VPA by cytochrome P450 (CYP450), which produces several metabolites, including 4-ene-valproic acid (4-eno-VPA) and 2,4-diene-valproic acid (2,4-diene-VPA) [9], which have been associated with an overproduction of ROS and the consequent loss of cellular homeostasis [16,17]. Regarding HO-AAVPA, it is distributed in various tissues, mainly in the liver, and it suffers biotransformation by the rat liver CYP2C11 isoform, producing hydroxylated metabolites [18,19,20].

Therefore, this study aimed to evaluate the liver toxicity of HO-AAVPA compared with VPA in a rat model by quantifying various biochemical parameters and performing macroscopic and histological analyses on in vivo-exposed liver tissue.

## 2. Results

### 2.1. Acute Toxicity

During the acute oral toxicity study of HO-AAVPA, there were no deaths after treatment. The signs of toxicity were not significant at the doses of 175 and 550 mg/kg; however, at 1750 mg/kg, acute dyspnea and lethargy were observed, while at the 2000 mg/kg dose, there was hyperactivity, piloerection, and dyspnea, culminating in a lethargic state. The animals behaved normally after 5 h of administration. The increase in body weight in both species (rats and mice) remained within the established parameters indicated by the growth curve compared with the control group (Table 1 and Table 2). Bulleted lists look like this:

### 2.2. Necropsy

The necropsy apparently did not show any macroscopic alteration of the organs and tissues compared with the control (Figure 1). Finally, the LD_50_ of HO-AAVPA was higher than 2000 mg/kg.

### 2.3. Effect of HO-AAVPA for 7 Days of Treatment at Repeated Doses

#### 2.3.1. Macroscopic Analysis

The liver of the animals that were administered with HO-AAVPA (708 mg/kg) for 7 days via i.p. in repeated doses was studied macroscopically, presenting a color change to intense red in some regions of the organ accompanied by an increase in size of the latter compared with the control, while with VPA (500 mg/kg, administered under the same conditions), an enlarged congestive appearance was observed, in addition to the presence of white granulomas or abscesses in the area (Figure 2).

#### 2.3.2. Effect of HO-AAVPA on Alanine Aminotransferase, Aspartate Aminotransferase, and Alkaline Phosphatase Activity

The ALT, AST, and ALP activity in the serum sample of the rats that were treated with HO-AAVPA remained close to that of the control, which, unlike the VPA group, presented a greater increase mainly in ALP activity; however, no significant difference was found in any of the cases compared with the control group (Table 3).

#### 2.3.3. Effect of HO-AAVPA on Lipoperoxidation and Reduced Glutathione Levels

The malondialdehyde (MDA) levels of the group treated with HO-AAVPA remained close to those of the control group, whereas these levels were higher in the group treated with VPA compared with the latter (Figure 3A). On the other hand, the GSH levels of the group treated with HO-AAVPA tended to recover with respect to the control group, while these levels in the group treated with VPA were much lower than the latter (Figure 3B).

#### 2.3.4. Glutathione Peroxidase (GPx) and Glutathione Reductase (GR)

Regarding the rat groups administered with HO-AAVPA and VPA, the GPx activity did not show a significant difference compared with the control group (Figure 4A). On the other hand, the GRx activity in the group treated with HO-AAVPA did not show any change with respect to the control; however, in the group with VPA, there was a tendency to decrease, but without presenting a significant difference (Figure 4B).

#### 2.3.5. Catalase Activity (CAT)

The CAT activity in the animals treated with HO-AAVPA presented an increase, while in the group treated with VPA there was a decrease, both groups compared with the control; however, in none of the cases was there a significant difference (Figure 5).

#### 2.3.6. Total Proteins

The quantification of the protein content in liver tissue showed a tendency to decrease in the group treated with VPA, although it was not statistically different in comparison with the group treated with HO-AAVPA (Figure 6); however, in none of the cases was there a significant difference with respect to the control group.

#### 2.3.7. Histopathological Examination of the Liver

On evaluation of the liver histology of rats, the control group did not reveal any apparent morphological alterations (Figure 7A). On the other hand, the liver samples of the HO-AAVPA group (Figure 7B) presented sinusoidal congestion and focal ballooning degeneration of hepatocytes, as well as infiltration; despite this, cell integrity is maintained in this group (Figure 7B). In the liver samples from the VPA group, there was hepatocyte infiltration (Figure 7C), in addition to ballooning degeneration that can be interpreted as disintegration of the cytoplasmic membrane, including eosinophilic changes, and complete dissolution of the hepatocyte nucleus (Figure 7D).

## 3. Discussion

Various studies have shown that HO-AAVPA has antiproliferative effects on different cancer cell lines [1,2,3]. Therefore, it is important to continue with the preclinical evaluation of this compound since the results found make it possible that, in the future, it could be used for the treatment of cancer. Therefore, the main aim of this work was to evaluate whether the compound HO-AAVPA produces toxicity both at a systemic and hepatic level, considering that drug-induced liver injury (DILI) remains the most common reason that drugs are not approved in clinical trials. Currently, DILI is considered the main factor in 50% of all cases of acute liver failure and accounts for 5% of all hospital admissions [21].

The evaluation of the acute oral toxicity test carried out based on the OECD Guide 425 [22] showed that all the animals administered with the dose of 2000 mg/kg of HO-AAVPA survived (Table 1 and Table 2), and, therefore, it is suggested that the LD_50_ value is above this dose, which, compared with that of VPA (LD_50_: 1465 mg/kg p.o.) in rats, is much higher [23,24]. Therefore, HO-AAVPA is classified in category IV according to the classification criteria for acute toxicity of the Globally Harmonized System of Classification and Labeling of Chemical Products (GSH, 300 < LD50 < 2000 mg/kg), which indicates that its oral administration is safe [25,26]. In addition, this result confirms what was previously reported (6). It is worth mentioning that at a dose of 2000 mg/kg the animals presented symptoms such as hyperactivity, piloerection, dyspnea, and a lethargic state, which suggests that at this dose the HO-AAVPA compound could interfere with the cholinergic or adrenergic systems as well as with the activity of the ɣ-aminobutyrate (GABA) neurotransmitter due to the fact that these symptoms also occur with VPA [15,27,28].

Likewise, during the acute oral toxicity study, it was shown that HO-AAVPA did not generate a significant body weight gain at the evaluated doses (175, 550, 1750, and 2000 mg/kg) in any of the species (Table 1 and Table 2). In this regard, reports show that VPA in humans induces an increase in body weight and sometimes requires discontinuation of treatment, as was the case in a study of epileptic patients in which it was recorded that 57% of 67 subjects gained weight, while the remaining 43% increased insignificantly; this effect seems to be due to a state of oxidative stress and increased adipose tissue. Previous reports suggest that the CYP2C19 polymorphism could be the cause of VPA-induced weight gain; however, more research is needed to verify these findings [14,18,29,30]. Therefore, this result would represent an advantage in relation to the HO-AAVPA compound, which would not have this side effect in patients. On the other hand, the macroscopic analysis of the organs of the animals administered with HO-AAVPA in the acute study revealed that apparently no damage was produced in them since the integrity of the tissues remained similar to that of the control group, even at the dose of 2000 mg/kg (Figure 1).

During the study for 7 days at repeated daily doses, it was possible to demonstrate that the liver damage caused by the group treated with VPA showed the presence of granules in addition to congested tissue, which, unlike HO-AAVPA, was found to a much lesser extent when compared with the control group (Figure 2). This suggests that no toxic metabolites are generated during the metabolism of HO-AAVPA [18,19] compared with VPA, as has been reported previously [9].

There are also some biochemical and toxicological parameters that can help diagnose whether HO-AAVPA induces hepatotoxicity [31]. Among the biochemical parameters that help to establish a diagnosis related to alterations in liver functions are the quantification of the activity of ALT, AST, and ALP [32]. In this sense, it was found that HO-AAVPA did not produce a significant change in the activity of these enzymes with respect to the control (Table 3); this suggests that the evaluated compound does not produce alterations at the liver level, unlike VPA, which produces an increase in ALT, AST, and ALP, suggesting a probable hepatotoxicity induced by this drug, as previously reported [11].

On the other hand, the levels of free radicals generated by a state of oxidative stress and loss of antioxidant capacity in the cell can be measured indirectly through the quantification of TBARS and GSH, as well as from the determination of the activity of the antioxidant enzymes GPx, GR, and CAT [9,33]. Regarding the determination of MDA levels, in the present study it was shown that HO-AAVPA induces a lower degree of lipid peroxidation than the VPA group (Figure 3), suggesting that HO-AAVPA, once metabolized, does not generate reactive intermediates that can lead to the formation of free radicals, but is only capable of producing hydroxylated compounds [18,19]. Regarding the GSH levels in the group treated with HO-AAVPA, these tended to decrease but not like those of the group treated with VPA (Figure 3), which suggests a greater hepatoprotective effect of HO-AAVPA, avoiding severe liver damage after repeated administration [20]. Under the above context, liver injury can be caused by an exogenous or endogenous agent that leads to oxidative stress due to the production of free radicals [34]. Despite the fact that cells have the ability to balance this oxidative process, the loss of cellular homeostasis can manifest itself in different ways, for example, through the metabolism of drugs that produce compounds with unpaired electrons that affect the expression of proteins related to free radical levels [35].

The GPx activity determined in the groups treated with HO-AAVPA and VPA did not show a significant difference with respect to the control (Figure 4). This correlates with the literature in which it is reported that GPx activity decreases when there is a considerable increase in the concentration of hydrogen peroxide (H_2_O_2_) and peroxyl (ROOH) produced in the different pathways of the antioxidant systems [35,36]. In the case of GRx activity, in the HO-AAVPA group, there was no change compared with the control group, unlike the VPA group, in which the activity was clearly decreased (Figure 4). These results show a close relationship with GSH levels, which in the HO-AAVPA group were close to those of the control, indicating that by maintaining GRx activity, GSH production is also favored. In this regard, it is well known that the endogenous antioxidant system is affected when there is a high production of ROS, leading to an alteration of GR activity to produce GSH from oxidized glutathione (GSSG) to prevent damage generated by oxidant molecules [34,35].

Related to CAT activity, the HO-AAVPA compound does not alter enzyme activity compared with the control (Figure 5), suggesting that H_2_O_2_ production is not increasing. However, the group treated with VPA showed a decrease in CAT activity, which means that this drug causes oxidative stress [15,36,37,38,39,40]. Finally, the results of the quantification of the protein content in liver tissue showed a tendency to decrease in the HO-AAVPA group with respect to the control group, but this was not so evident in comparison with the VPA group, which, according to what was reported [16,40,41], presents a significant decrease in total protein levels [41]. This suggests that HO-AAVPA does not significantly damage liver tissue.

The histopathological studies of the treatment for 7 days at repeated doses daily showed that the group treated with HO-AAVPA presented mainly infiltration, unlike the group treated with VPA (Figure 7). However, VPA induces severe liver damage, including infiltration of hepatocytes, ballooning degeneration that includes disintegration of the cytoplasmic membrane with eosinophilic changes, and complete dissolution of the nucleus of hepatocytes. It agrees with other reports regarding chronic evaluations with VPA producing inflammation, necrosis, and congestion, which may be due to a state of oxidative stress and the generation of toxic metabolites (4-ene-VPA and 2,4-diene-VPA) during their biotransformation by CYP450 [15,16,17,42,43]. Instead, it has been suggested that the biotransformation of HO-AAVPA generates hydrophilic products (hydroxylated metabolites), reducing side effects [19,20] which have antiproliferative effects on breast cancer cells [39]. In addition to the above, in vivo studies found that in a short time, HO-AAVPA has a good distribution among tissues, mainly in the liver, which suggests that this compound could be easily excreted, thus avoiding the production of hepatic damage [18]. These results lead to the fact that the HO-AAVPA compound produces alterations in hepatic histoarchitecture to a lesser degree than VPA, as has been previously reported [11,39].

In summary, these results suggest that the compound HO-AAVPA can act as a free radical scavenger, thus avoiding liver damage that could be generated by repeated administrations, as has been reported in the case of several drugs, including VPA [44,45]. This correlates with a previous study in which HO-AAVPA was shown to exhibit antioxidant capacity in vitro by the 2,2-diphenyl-1-picryl-hydrazyl-hydrate (DPPH) radical test, suggesting that this activity is due to the presence of the aromatic ring together with the amide bond. In addition, it is important to highlight that HO-AAVPA induces less hepatotoxicity compared with other anticancer drugs that have even been administered at lower doses and times by i.p. in rat models [44,46,46,47,48,49,50,51].

## 4. Materials and Methods

### 4.1. Reagents and Equipment

All chemicals were analytical grade and purchased by Sigma Aldrich (Toluca, State of Mexico, Mexico). A batch of HO-AAVPA [1,2], patent application MX/a/2013/002766) was synthesized in the New Drug Design and Development and Biotechnological Innovation laboratory with a purity of 99.2% [18,19]. The kits for the quantification of biochemical parameters were purchased by Randox México, S.A. de C.V. (Tlalneplantla, State of Mexico, and Mexico).

### 4.2. Experimental Animals

Male and female Wistar rats (180 ± 20 g) and male and female CD1 mice (25 ± 5 g) were purchased from the Universidad Autónoma Metropolitana Unidad Xochimilco, Mexico City, and housed in polypropylene cages under controlled temperature conditions (20–25 °C) and light/dark cycles of 12 × 12 h, with food (standard) and water on free demand. Before carrying out the experiments, the animals were adapted to their new habitat during an acclimatization period of one week. Animal procedures were performed in accordance with the guidelines of the Declaration of Helsinki and the Official Mexican Standard [52,53]. Technical Specifications for the Production, Care and Use of Laboratory Animals, SAGARPA, as well as the “Guide for the care and use of laboratory animals” of the National Research Council and Institutes of Health (NIH Publications No. 8023, revised in 1978). In addition, the animal protocol was approved by the Research Committee for the Care and Use of Laboratory Animals (CICUAL) of the Escuela Superior de Medicina-IPN (Approval number: ESM.CICUAL-03/25-08-2015). At the end of each experiment, all animals were handled and euthanized according to humane endpoint considerations [45,54].

### 4.3. Acute Toxicity Evaluation

Thirteen rats of the Wistar strain (females and males; 180 ± 20 g) and thirteen mice of the CD1 strain (females and males; 25 ± 5 g) were used to determine the LD_50_ value using the Up-and-Down method. The first animal receives a recommended dose of 175 mg/kg, if the animal survives, the dose for the next animal is increased by a progression factor of 3.2 times the original dose; if it dies, the dose for the next animal is reduced by a modified dose with the same progression factor according to Guide No. 425 of the Organization for Economic Cooperation and Development [22]. Before starting the administration, the animals were fasted for a period of 12 h (rats) and 6 h (mice), starting with the group of female rats, which were rigorously observed for 48 h and up to 14 days. Since all previously treated rats survived, three other male animals were administered doses of 2000 mg/kg, which survived the study period (14 days). Subsequently, the same protocol was repeated in CD1 mice (females and males), and all animals were constantly observed for 14 days. The body weight of each rat was recorded weekly. Animals were sacrificed with 72 mg/kg of sodium pentobarbital intraperitoneally (i.p.), and the peritoneum was exposed for macroscopic analysis of the organs. Subsequently, the liver was perfused with 0.1 M phosphate buffered saline (PBS) and pH 7.4, cut, and removed to be immediately placed on ice and available for the corresponding histopathological study. The LD_50_ value was estimated using the software indicated in OECD Guide 425 [22].

### 4.4. Evaluation of HO-AAVPA for 7 Days

#### 4.4.1. Animals

Twenty-eight male Wistar rats were used to carry out the repeated doses study, which was divided into four groups (Table 4) (n = 7). The dose to be evaluated of HO-AAVPA (708 mg/kg per day) was calculated in equimolar relation to the hepatotoxic dose of VPA (sodium valproate, 500 mg/kg per day) [12,23,28,43,44,55]. It should be mentioned that in the case of groups 3 and 4, the respective doses were divided equally into two administrations per day (morning and afternoon), with an interval of 12 h between them. The treatment lasted 7 days, and the administration was via i.p. The animals were rigorously observed after the treatments for 2 h, since this is the time in which the parent compound is absorbed from 1 to 2 h [56]. Finally, these were sacrificed with sodium pentobarbital (72 mg/kg, i.p.).

#### 4.4.2. Determination of Liver Damage by Macroscopic Analysis

The sternum region was exposed to carry out the corresponding macroscopic analysis; later, the blood samples were collected by cardiac puncture and immediately centrifuged to obtain the serum that was used in the biochemical studies. The liver was perfused with 0.1 M PBS and pH 7.4, dissected, and fixed in formaldehyde to be used in histological studies. Other portions of the tissue were stored at −20 °C for subsequent evaluations of biochemical parameters.

#### 4.4.3. Measurement of Alanine Aminotransferase and Aspartate Aminotransferase Activity

Liver damage was determined using a serum sample from each rat to quantify ALT and AST activity using the ALT1268 and AST1267 kits (Randox México, S.A. de C.V.), respectively. In both cases, enzyme activity was determined by oxidation of NADH to NAD+ at 365 nm using an ultraviolet-visible (UV-Vis) spectrophotometer (PerkinElmer Lambda 25 UV-Vis-Spectrophotometer (Waltham, Massachusetts 02451, U.S.A.)). The activity was expressed in U/L.

#### 4.4.4. Measurement of Alkaline Phosphatase Activity

ALP activity was quantified with a serum sample from each rat using the AP307 kit (Randox México, S.A. de C.V.), the substrate being p-nitrophenyl phosphate, which was measured at 405 nm in a spectrophotometer. UV-Vis (PerkinElmer Lambda 25 UV-Vis Spectrophotometer (Waltham, Massachusetts 02451, U.S.A.)). The enzymatic activity was expressed in U/L.

#### 4.4.5. Determination of Malondialdehyde Levels

The content of TBARS was determined indirectly in the liver based on what was previously reported with some modifications [57,58]. Each liver tissue sample (50 mg) was homogenized in 500 µL of distilled water, then 350 µL of 0.15 M Tris-HCl buffer, pH 7.4, were added and incubated at 37 °C for 30 min. Subsequently, 1 mL of 0.375% (p/v) thiobarbituric acid dissolved in 15% (p/v) trichloroacetic acid was added, brought to a boil for 1 h, then cooled and centrifuged at 8000 rpm for 15 min; then the supernatants were extracted and read at 540 nm by means of a UV-Vis spectrophotometer (UV-Vis-PerkinElmer Lambda 25-Spectrophotometer (Waltham, MA 02451, USA)). In the blank, the supernatant was replaced by distilled water. 1,1,3,3-tetramethoxypropane (TMP) was used as a commercial standard. Finally, the MDA content was expressed as nmol/mg of tissue.

#### 4.4.6. Measurement of Reduced Glutathione Levels

GSH content was determined as previously reported with some modifications [59]. Each liver tissue sample (50 mg) was homogenized with metaphosphoric acid (2.5 mL, 3% (*w*/*v*)), then centrifuged at 3000 rpm for 15 min. An aliquot of supernatant (250 µL) was added to a test tube containing 250 µL of 0.1 M phosphate buffer, pH 8, followed by 10 µL of 5,5′-dithiobis-2-nitrobenzoic acid (DTNB), which were mixed and read at 415 nm using a UV-Vis spectrophotometer (PerkinElmer Lambda 25 UV-Vis Spectrophotometer (Waltham, Massachusetts 02451, U.S.A.)). In the blank, the supernatant was replaced by 3% (*w*/*v*) metaphosphoric acid. The standard used was commercial GSH. GSH content was expressed as nmol/mg tissue.

#### 4.4.7. Determination of Glutathione Peroxidase and Glutathione Reductase Activity

GPx and glutathione reductase (GR) activity were quantified in rat serum using the RS504 and GR2368 kits (Randox México, S.A. de C.V.), respectively. GPx activity was determined by oxidation of NADPH to NADP+ at 320 nm. Likewise, GR activity was determined by the reduction of oxidized glutathione (GSSG) in the presence of NADPH, which is oxidized to NADP+ at 340 nm [59]. Both activities were measured on a UV-Vis spectrophotometer (PerkinElmer Lambda 25-UV-Vis Spectrophotometer (Waltham, MA 02451, USA)). The results were expressed in U/L.

#### 4.4.8. Measurement of Catalase Activity

CAT activity was quantified in serum using kit No. 707,002 (Cayman Chemical). The method is based on the measurement of the decrease in hydrogen peroxide (H_2_O_2_) consumption, which was read at 540 nm using an Elisa reader (Thermo Fisher Scientific Multiskan EX-100 (Thermo Fisher Scientific Oy P.O. Box 100, FI-01621 Vantaa, Finland)). The activity was expressed in nmol/min/mL.

#### 4.4.9. Protein Quantification

Protein determination was performed using kit No. 704,002 (Cayman Chemical, Mexico) based on the Bradford method. The absorbance was measured at 595 nm in an Elisa reader (Thermo Fisher Scientific Multiskan EX-100 (Thermo Fisher Scientific Oy P.O. Box 100, FI-01621 Vantaa, Finland)). The results were expressed in mg of protein/mL.

#### 4.4.10. Histopathological Studies

Liver samples from the experimental and control groups were processed, cleaned, and placed in 0.9% (*w*/*v*) NaCl physiological solution for 30 min, then fixed in 10% formaldehyde solution for 24 h. The processed tissues were dehydrated in increasing concentrations of 70%, 80%, 90%, 96% absolute alcohol, and finally with xylol for 1 h for the substitution of alcohol by xylol. Then the tissues were submerged in molten paraffin (62 °C) for the xylol to come out of the tissue and be replaced by paraffin. Finally, cubic molds containing the tissue were filled with paraffin, and after cooling, they were cut with the aid of a sliding microtome (rotary hand processor STP 120, Thermo Scientific MICROM, International GmbH (Robert-Bosch-Str. 49 69,190 Walldorf/Germany) in sections in serial layers with a thickness of 4 mm. These were extended by floating on the surface of warm water. Subsequently, the slides were immersed in descending concentrations of xylol and alcohol for tissue staining by the hematoxylin method and eosin (H and E). This process was performed in triplicate to obtain adequate image resolution of the tissue. Finally, examination was conducted through the light electron microscope at 20X [11,23,38].

### 4.5. Statistical Analysis

All data are presented as the mean ± standard error of the mean (±SEM). A multiple sample comparison was applied to determine if there was a significant difference between the groups using ANOVA and Tukey’s multiple range test as a post-hoc test. Statistical significance was considered with a *p*-value < 0.05.

## 5. Conclusions

Based on the results obtained in this study, it can be concluded that the HO-AAVPA compound does not induce systemic or hepatic toxicity, which was demonstrated by quantifying various biochemical parameters. Therefore, this suggests that this compound does not interfere with cellular homeostasis since it maintains the molecular integrity of tissues by reducing or preventing the oxidative stress that is generated by VPA.

## Figures and Tables

**Figure 1 molecules-28-06282-f001:**
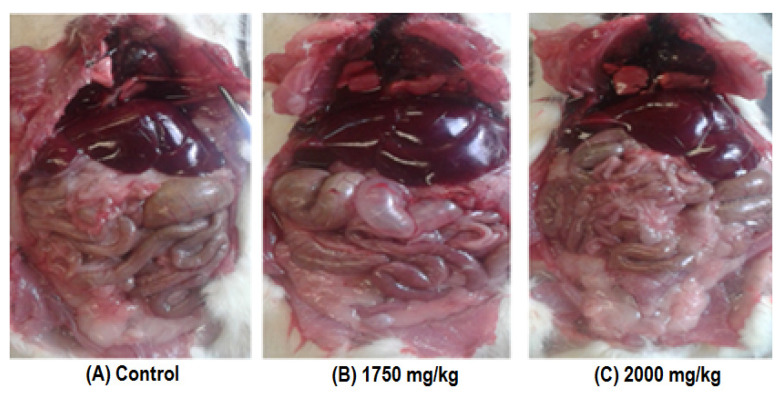
Macroscopic analysis of the organs and tissues of the animals administered with HO-AAVPA in the acute oral toxicity study.

**Figure 2 molecules-28-06282-f002:**
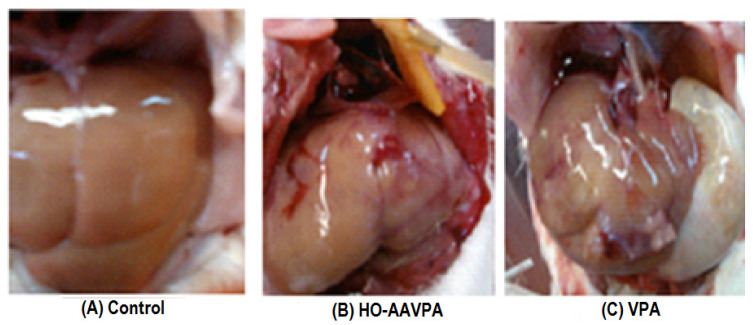
Macroscopic analysis of the liver of animals administered HO-AAVPA (708 mg/kg) and VPA (500 mg/kg) for 7 days at repeated doses applied daily.

**Figure 3 molecules-28-06282-f003:**
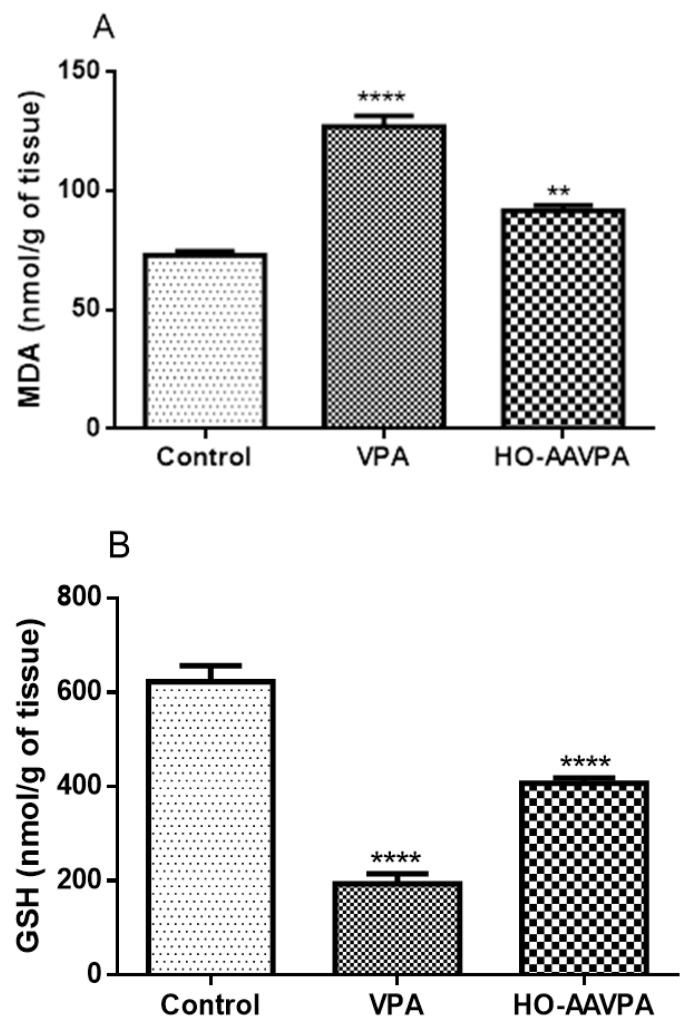
Effect of HO-AAVPA on MDA (**A**) and GSH (**B**) levels in rat liver samples. Data are represented as the mean (M) ± SEM, using one-way ANOVA and Tukey as a post-hoc test. MDA: ** HO-AAVPA vs. control, **** VPA vs. control (n = 7; *p <* 0.05); GSH: **** HO-AAVPA or VPA vs. control (n = 7; *p* < 0.05).

**Figure 4 molecules-28-06282-f004:**
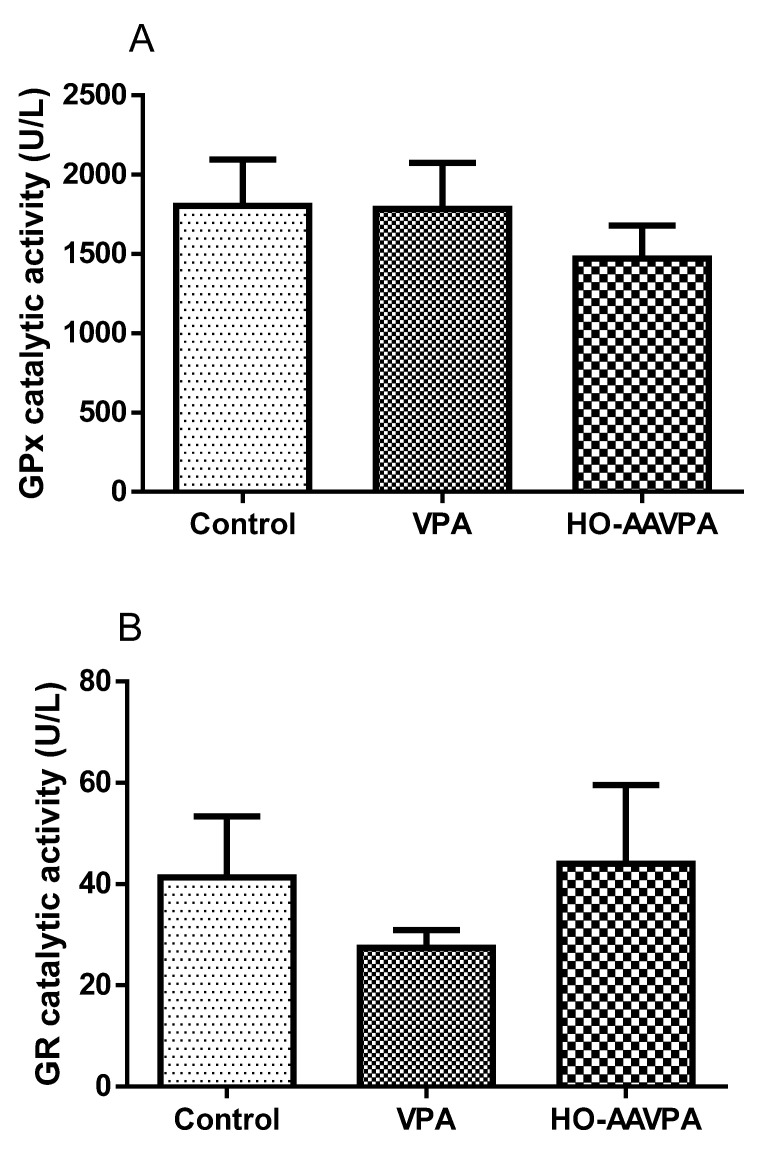
Effect of HO-AAVPA on GPx (**A**) and GR (**B**) activity in rat serum. Data are represented as the mean (M) ± SEM, using one-way ANOVA and Tukey as a post-hoc test. GPx: HO-AAVPA or VPA vs. control, and GR: HO-AAVPA or VPA vs. control. No statistically significant difference (n = 7; *p* < 0.05).

**Figure 5 molecules-28-06282-f005:**
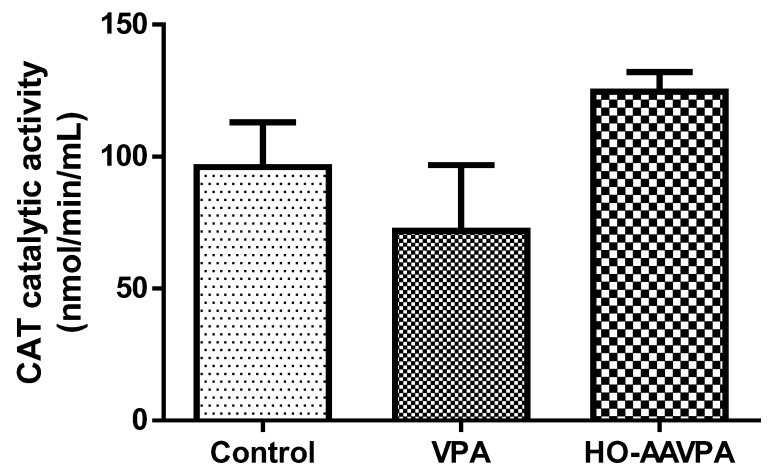
Effect of HO-AAVPA on CAT activity in rat serum samples. Data are represented as the mean (M) ± SEM, using one-way ANOVA and Tukey as a post-hoc test. CAT: HO-AAVPA or VPA vs. control: No statistically significant difference (n = 7; *p <* 0.05).

**Figure 6 molecules-28-06282-f006:**
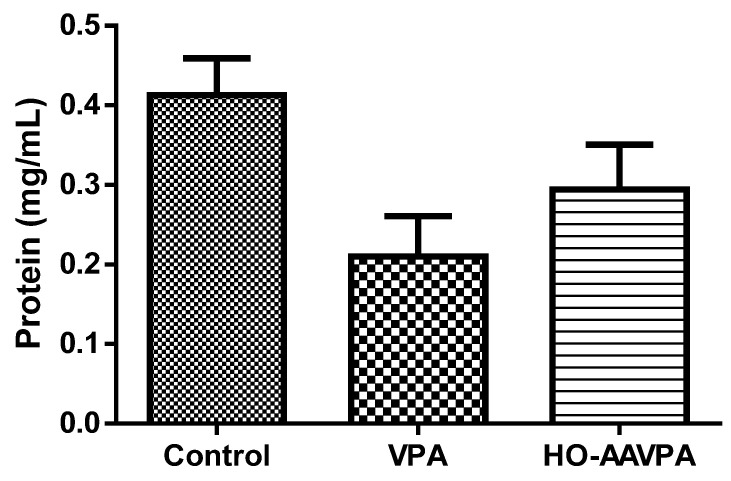
Effect of HO-AAVPA on protein levels in rat liver samples. Data are represented as the mean (M) ± SEM, using one-way ANOVA and Tukey as a post-hoc test. HO-AAVPA or VPA vs. control (n = 7; *p <* 0.05).

**Figure 7 molecules-28-06282-f007:**
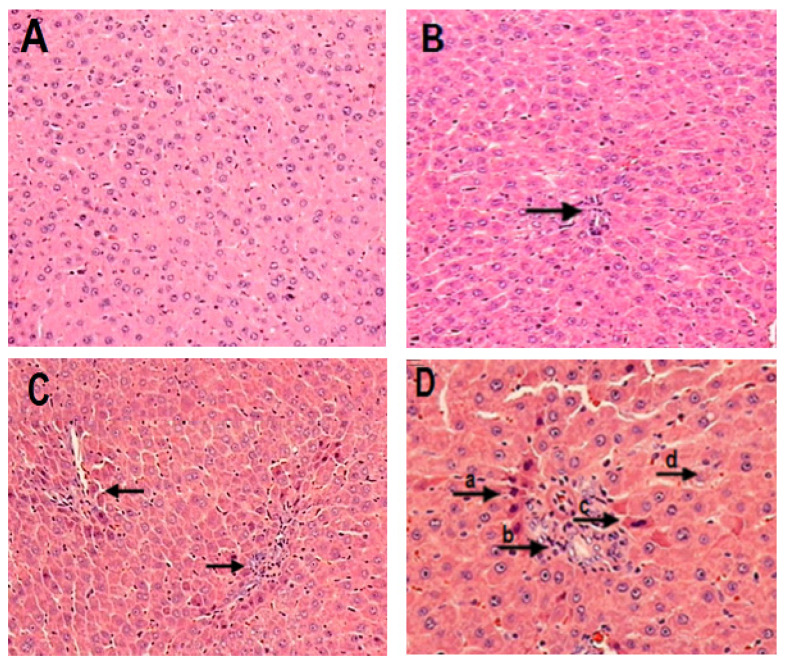
Representative microphotographs of liver tissue stained with H and E at 20× of: (**A**) Control group without morphological alterations in the hepatocytes. (**B**) HO-AAVPA (black arrow): sinusoidal congestion and focal ballooning degeneration of hepatocytes. (**C**) VPA (black arrows): infiltration. (**D**) VPA (black arrows) (c and b): infiltration, and (a, b, and d): ballooning degeneration of hepatocytes that includes disintegration of the cytoplasmic membrane and complete dissolution of the nucleus, as well as sinusoidal congestion.

**Table 1 molecules-28-06282-t001:** Body weight of Wistar strain rats treated with HO-AAVPA during the acute oral toxicity test.

Treatment Dose (mg/kg)		Weight (g)		
		0 Day	7 Days	14 Days
Female	175	193	220	226
	550	202	229	231
	1750	199	221	231.5
	2000	203	230	271
		205	225	273
		217	234	285
		239	272	324
	Control	209	227	231.50
	Vehicle	195	216	222
Male	2000	189.2	237.5	291
		199.2	248	301
		195.2	241	292
		193.2	241	291
		193.7	242	299.5
		198.7	246	300
	Control	203	243	293.5
	Vehicle	198.7	241	277

**Table 2 molecules-28-06282-t002:** Body weight of CD1 mice treated with HO-AAVPA during the acute oral toxicity test.

Treatment Dose (mg/kg)		Weight (g)		
		0 Day	7 Days	14 Days
Female	2000	22.7	29	40.5
		23.7	29.5	32
		24.7	28	31
		24	26	27
		27.57	29	32
		23	24	27
	Control	26	28.5	31
	Vehicle	24.5	27	29.5
Male	2000	33	35	36.5
		34.2	39.5	42
		32	34	36
		34	35	38
		30.5	31	32.5
		29.5	31.5	34
	Control	33	35	36.5
	Vehicle	29.5	30.5	32

**Table 3 molecules-28-06282-t003:** Effect of HO-AAVPA on the activity of enzymes related to liver damage (n = 7).

Treatment	ALT (U/L)	AST (U/L)	ALP (U/L)
Control	25.43 ± 4.69	54.93 ± 6.47	63.87 ± 16.97
VPA (500 mg/kg)	41.90 ± 12.46	63.05 ± 4.58	82.59 ± 12.06
HO-AAVPA (708 mg/kg)	23.78 ± 4.96	57.86 ± 8.28	75.25 ± 8.58

Abbreviations: ALT: alanine aminotransferase, AST: aspartate aminotransferase, ALP: alkaline phosphatase, VPA: valproic acid, and HO-AAVPA: *N*-(2-hydroxyphenyl)-2-propylpentanamide. Data are represented as the mean (M) ± SEM, using one-way ANOVA and Tukey as a post-hoc test. HO-AAVPA vs. control: No statistically significant difference (n = 7; *p* < 0.05).

**Table 4 molecules-28-06282-t004:** Random distribution of the animals in the different study groups (n = 7, N = 28).

Group	Treatment
G1: Control	without treatment
G2: Vehicle	propylene glycol40%, Tween 80 5%, and y Saline solution 55% (1 mL/kg)
G3: VPA	500 mg/kg
G4: HO-AAVPA	708 mg/kg (Equimolar relationship with respect to VPA)

Abbreviations: VPA: Valproic acid (sodium valproate); HO-AAVPA (*N*-(2-hydroxyphenyl)-2-propylpentanamide).

## Data Availability

All the relevant data found in the study are available in the article.

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
