# Peer review of "Hepatotoxic Evaluation of N-(2-Hydroxyphenyl)-2-Propylpentanamide: A Novel Derivative of Valproic Acid for the Treatment of Cancer"

_molecules, 2023, doi:10.3390/molecules28176282_

Round 1
Reviewer 1 Report
The manuscript evaluated the liver injury of HO-AAVPA,a recently designed VPA derivative.
There are major concerns needed to be addressed.
1.The compound HO-AAVPA was synthesized in the New Drug Design and Development and Biotechnological Innovation laboratory. How about the purity of the compound? What are the basic physical and chemical properties? It is suggested to provide the relevant pharmacodynamic evaluation data.
2. The evaluation of HO-AAVPA was carried out for 7 days. Why did the authors set the study time at 7 days? It seems that 7 days is not long enough for the conclusion.
The language needs to be properly polished.
Author Response
https://drive.google.com/drive/folders/10K6bi05QwIBHVzpbApKa--C6RziQ0yRN?usp=drive_link

Reviewer 2 Report
The most significant side effect of drugs such as anticancer drugs is hepatotoxicity, so I think comparing hepatotoxicity to existing drugs is a good idea. This study would be used as important information to secure the safety of HO-AAVPA as an anticancer drug in the future. Moreover, this manuscript has been written to be well understood. However, I bring some following comments to the authors' attention.
1. The total number of animals used in the oral toxicity test was presented, but it is necessary to present the number for each administration group.
2. Description of the dosage setting of 175, 550, and 1750 mg/kg used in oral toxicity testing is required.
3. Evaluating organs collected following oral toxicity and repeated toxicity studies with only macroscopic analysis is thought to be insufficient to evaluate toxicity. If toxicity is caused by drugs, an increase or decrease in organ weight could be occurred compared to the control group. Why don't you add supporting data such as organ index?
4. In the repeated toxicity test, it seems that additional explanations are needed on the basis of the decision to use ip injection, not oral administration.
5. Line 210, LD50 obtained based on oral study was compared with LD50 calculated based on i.v. or i.p. injection study. Since oral administration shows generally less toxic than direct administration into the blood (i.v. or i.p.), it cannot be compared with the results obtained by oral administration. In particular, since this paper explains that it is a lower toxic derivative through a toxicity comparison with VPA, the discussion of the present part should be needed to be rewritten by comparing it with other data.
6. In line 337, all animals have 12 hours of fasting time, do the mouse and rat have the same fasting time? The mouse usually takes 3-4 hours of fasting time, so please check if the description is correct.
7. Line 350, in general, subchronic study means to observe toxicity effects after repeated administration for 3 months (at least 1 month), so the word 'subchronic study' needs to be corrected for readers' understanding.
8. Please check the typos, style of the table, and etc.
- Line 57, the dash should be eliminated for the word ‘inhibition’.
- Line 71, micceles should be corrected.
- Alignments of the titles (Female, Male) in Table 1 are required.

Overall, the documentation was well written to understand.
Author Response
https://drive.google.com/drive/folders/1_n6N6f09mU0eYjd0e5BRCcKTGKaZ6Nit?usp=sharing

Round 2
Reviewer 1 Report
We still think that the evaluation time of HO-AAVPA for 7 days is not long enough for the conclusion. As a new compound, the basic physical and chemical properties and the toxicity seems lower than VPA. So, it is not enough to refer to VPA only.
no
Author Response
https://drive.google.com/drive/folders/1gxotEDS47rStO9xEtHhkbeMrWGyxMfk9?usp=drive_link
